# Sexual selection theory meets disease vector control: Testing harmonic convergence as a "good genes" signal in *Aedes aegypti* mosquitoes

**Garrett P. League**[1], **Laura C. Harrington**[1], **Sylvie A. Pitcher**[1], **Julie K. Geyer**[1¤], **Lindsay L. Baxter**[1], **Julian Montijo**[1], **John G. Rowland**[2], **Lynn M. Johnson**[3], **Courtney C. Murdock**[1,4,5], **Lauren J. Cator**[2]*

**1** Department of Entomology, Cornell University, Ithaca, New York, United States of America, **2** Department of Life Sciences, Imperial College London, Silwood Park, Ascot, United Kingdom, **3** Cornell Statistical Consulting Unit, Cornell University, Ithaca, New York, United States of America, **4** Department of Infectious Diseases, University of Georgia, Athens, Georgia, United States of America, **5** Odum School of Ecology, University of Georgia, Athens, Georgia, United States of America

¤ Current address: Department of Biology, University of North Carolina at Chapel Hill, Chapel Hill, North Carolina, United States of America
* l.cator@imperial.ac.uk

**Data Availability Statement:** All relevant data are within the manuscript and its Supporting Information files.

## Abstract

### Background

The mosquito *Aedes aegypti* is a medically important, globally distributed vector of the viruses that cause dengue, yellow fever, chikungunya, and Zika. Although reproduction and mate choice are key components of vector population dynamics and control, our understanding of the mechanisms of sexual selection in mosquitoes remains poor. In "good genes" models of sexual selection, females use male cues as an indicator of both mate and offspring genetic quality. Recent studies in *Ae. aegypti* provide evidence that male wingbeats may signal aspects of offspring quality and performance during mate selection in a process known as harmonic convergence. However, the extent to which harmonic convergence may signal overall inherent quality of mates and their offspring remains unknown.

### Methodology/Principal findings

To examine this, we measured the relationship between acoustic signaling and a broad panel of parent and offspring fitness traits in two generations of field-derived *Ae. aegypti* originating from dengue-endemic field sites in Thailand. Our data show that in this population of mosquitoes, harmonic convergence does not signal male fertility, female fecundity, or male flight performance traits, which despite displaying robust variability in both parents and their offspring were only weakly heritable.

**Funding:** This work was funded by the National Institute of Allergy and Infectious Diseases (www.niaid.nih.gov) grant R21AI118593 awarded to LJC and R01AI095491 awarded to LCH. The funders had no role in study design, data collection and analysis, decision to publish, or preparation of the manuscript.

**Competing interests:** The authors have declared that no competing interests exist.

## Conclusions/Significance

Together, our findings suggest that vector reproductive control programs should treat harmonic convergence as an indicator of some, but not all aspects of inherent quality, and that sexual selection likely affects *Ae. aegypti* in a trait-, population-, and environment-dependent manner.

### Author summary

Mosquitoes transmit numerous pathogens that disproportionately impact developing countries. The mosquito *Aedes aegypti*, studied here, transmits viruses that cause neglected tropical diseases such as dengue, yellow fever, chikungunya, and Zika. Disease prevention programs rely heavily upon mosquito vector control. To successfully interrupt disease transmission, several control methods depend upon the ability of laboratory-modified male mosquitoes to successfully mate with wild females to suppress or replace natural populations. However, our understanding of what determines mating success in mosquitoes is far from complete. Our study addresses the question of whether female *Ae. aegypti* mosquitoes use male acoustic signals to select higher quality mates and improve their offspring's fitness. We find that acoustic signals do not serve as universal indicators of fitness. Further, the fitness metrics we measured were only weakly heritable, suggesting that females that mate with high quality males do not necessarily produce fitter offspring. Our study provides a nuanced understanding of mate choice, mating acoustic signals, and parent and offspring reproductive fitness in a key disease-transmitting mosquito species. These discoveries improve our grasp of sexual selection in mosquitoes and can be leveraged by the vector control community to improve vitally important disease prevention programs.

## Introduction

*Aedes* mosquitoes are primarily responsible for the estimated 390 million annual dengue fever virus infections [1] as well as the millions of infections involved in recent yellow fever epidemics in Brazil [2], and chikungunya and Zika outbreaks in the Americas and the Caribbean [3]. The historically unprecedented distribution and expansion of *Aedes* mosquito populations and the diseases they spread ensure that vector control will remain a top public health priority in the 21st century and beyond [4–6]. Furthermore, as many current vector control programs rely primarily on only two classes of insecticides, and all four major insecticide classes are facing widespread resistance in *Aedes* populations across the globe [7], novel vector control methods are urgently needed.

Several recently developed control strategies involve mass laboratory rearing of modified mosquitoes that are released into nature to reduce or replace mosquito populations and disrupt disease transmission [8–11]. Critically, the success of such methods depends upon the ability of modified mosquitoes to successfully mate with their natural counterparts. Both modification itself as well as mass-rearing can affect mosquito mating success and compromise the biological efficacy and economic feasibility of otherwise promising disease control programs [12–14]. However, critical gaps in our present understanding of mosquito sexual selection and mating success limit our ability to fully leverage mosquito mating biology for effective vector control [15].

Mating in *Aedes* mosquitoes is rapid (<60 s) [16] and most often occurs in aerial swarms composed primarily of males and only a small number of females, but also in small groups or individually [17,18]. These highly skewed swarm sex ratios, coupled with robust female rejection behaviors toward potential mates [19–22] suggest an important role for female mate choice in this mating system, which was previously considered to be random [18]. Indeed, despite their brevity, mosquito mating interactions involve a complex set of multimodal behaviors both before, during, and after copulation [20,23] that include olfactory [24], visual [25], tactile [20], and, importantly, acoustic [26,27] stimuli. Males and females that come into close proximity during courtship flight prior to mating alter their flight tones to harmonize at various harmonic components in a phenomenon known as harmonic convergence [26]. As harmonic convergence in *Ae. aegypti* correlates with mating success [19,20,28], previous studies have suggested that females may use male acoustic cues to distinguish between potential mates [20,23,26].

Studies across a wide range of animal taxa have documented female preference for males with particular characteristics [29]. In some cases, variation in direct material benefits offered by males to females, such as nutritional gifts or parental effort, underlie these preferences, although no examples of such direct benefits have been documented in mosquitoes [17,30,31]. In other cases, males appear to offer only their genes [29,32]. However, the precise nature of the selection pressures that lead to female preference based on variation in genetic, or indirect benefits remains controversial [33,34]. There are two main models of indirect sexual selection [35]: good genes models [36,37] and Fisherian, or "sexy sons" models [38,39]. One key difference between these models concerns the nature of the male traits that females select: in good genes models, females select male traits that are intrinsically tied to inherent quality, whereas in Fisherian models, male traits are not indicative of inherent quality and will be selected for if they increase male mating success, regardless of other costs. However, these models are not mutually exclusive, with both predicting heritable correlations between sexually selected traits and offspring fitness [35]. In support of good genes models, a recent study in *Ae. aegypti* showed that parental harmonic convergence correlates with aspects of offspring immune performance [40]. Consistent with Fisherian predictions, another study in *Ae. aegypti* showed that male offspring resulting from harmonically converged parents are themselves more likely to converge and copulate successfully in competitive mating interactions [19]. However, as no other parent or offspring traits were measured in these studies, the extent to which harmonic convergence is indicative of overall heritable male quality remains unknown.

Here, we tested two key predictions of indirect sexual selection in a population *of Ae. aegypti* recently collected from the field using an expanded panel of male fertility, female fecundity, and male flight performance measures. First, we tested the extent to which harmonic convergence functions as a good genes signal of overall inherent quality in both parents and their offspring. Second, to test whether females that mate with higher quality males produce higher quality offspring, we assessed the standing variation and heritability of these traits in both parents and offspring. We found no evidence that harmonic convergence serves as a good genes signal of inherent quality for the fitness traits measured here. Further, we found that the male and female fitness traits examined here were variable, but only weakly heritable. These experiments highlight the importance of testing models of sexual selection in mosquitoes to better inform modified mosquito mass-release control strategies.

## Methods

### Experimental design

We examined the potential for harmonic convergence to signal heritable genetic quality across several key fitness traits (Fig 1). To investigate the potential of harmonic convergence as an

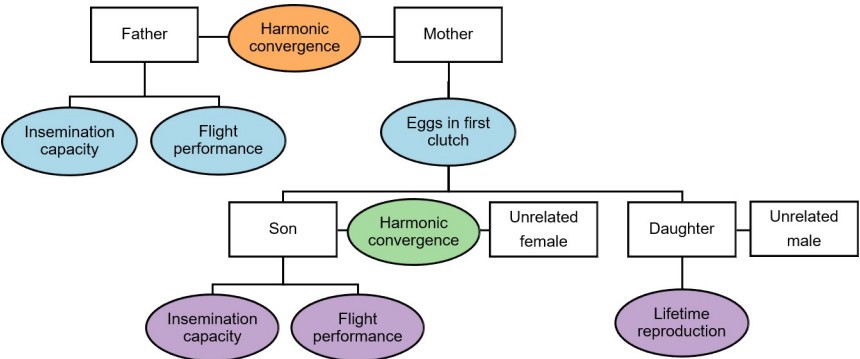

**Fig 1. Overview of experiments.** Flowchart illustrating the overall experimental design. Parental pairs were first tested for harmonic convergence. Father fitness was then assayed in either insemination capacity or flight performance assays and mother fitness was measured by first egg clutch size. Finally, sons were tested for harmonic convergence with unrelated females, followed by either insemination capacity or flight performance assays, while daughters were mated with unrelated males and their lifetime reproductive fitness tracked using a life table analysis.

indicator of inherent fitness, we first recorded aerial mating interactions of individual male-female parental pairs and determined their harmonic convergence status [19,20]. Parental pairs and their offspring were maintained separately to examine traits previously used to assess reproductive fitness in this species, including father and son insemination capacity [41], mother first egg clutch size [42], and daughter lifetime egg production and daily survival [30]. In a parallel set of experiments examining the signaling and heritability of male flight performance, we measured parental harmonic convergence outcomes and tested fathers and sons in a novel flight performance assay.

## Parental rearing

All experiments were performed across two generations ($F_2$ parents and their offspring for experiment one and $F_3$ parents and their offspring for experiment two) using a low generation (i.e., five or less generations in the laboratory [43]) *Ae. aegypti* population originating from a single collection of eggs from Kamphaeng Phet Province, Thailand (16˚27′ 48"N, 99˚31′ 47"E). Mosquito rearing was performed based on previously described methods to obtain medium-body-sized adults with size measured by proxy using wing lengths [30,41,44] (for details, see S1 Text). All experiments were conducted on mosquito adults at 2–6 days post-emergence, as at this age range, related Thai females have reached reproductive maturity and display the full range of feeding and host-seeking behaviors [30].

## Harmonic convergence assay

All parental recordings were performed between the hours of 09:00–18:00 and the temperature and humidity conditions were 26.47 ± 1.13˚C and 59.01 ± 7.95% RH for the reproductive fitness assays and 25 ± 1.8˚C and 68.1 ± 2.0% RH for the flight performance assays. For audio recording of potential instances of harmonic convergence, females were tethered and recorded as previously described [23] (S1 Text). For testing harmonic convergence prior to reproductive fitness assays, three virgin males were added simultaneously into the arena upon initiation of female flight and for testing convergence before flight performance assays, a single focal male was presented. For both assays, males and females were given five minutes to court before starting a new trial with different individuals. The first instance of a male-female courtship flight interaction was recorded using Audacity 2.1.3 software (https://www.audacityteam.org/)

and Raven Pro 1.5 software [45] (Bioacoustics Research Program, Cornell Laboratory of Ornithology, Ithaca, NY, USA). The recording was terminated either after the female rejected the male or a mating copula was formed, as described previously [19].

To determine whether male and female wing beat harmonics converged during flight interactions, audio files were analyzed as spectrograms using Raven Pro software as previously described [23] (S1 Text). The experiment was performed twice for both reproductive fitness (119 total mating pairs: 59 pairs recorded in experiment one and 60 pairs in experiment two) and male flight performance experiments (84 total pairs: 42 pairs recorded in experiments one and two). Only those pairs for whom a definitive converged or non-converged outcome could be determined based on our criteria were used in subsequent analyses (see final samples sizes in Results section).

## Offspring generation and rearing

Upon completion of a harmonic convergence assay flight recording, the interacting male was removed from the recording arena and the female was removed from her tether. The couple was then placed in a 0.5 L cardboard cup to form a parental mating pair for the generation offspring of known parental convergence status. To ensure that all pairs had mated successfully during the parental mating period, couples were held overnight and supplied with 10% sugar-soaked pads to ensure ample time and resources for mating and survival. After this period, females were transferred to individual containers and provided access to a human host (co-authors SAP or LJC) until females had fed to repletion, which was verified by visual examination for engorgement (Cornell IRB Human Subjects Activity Exemption, FWA 00004513, and Imperial College London Health and Safety approved). Gravid females were provided oviposition sites for six days post-blood meal. Their first clutch of eggs was then counted and labelled with an individual ID to enable offspring tracking by parental convergence status in subsequent experiments. Offspring from each pair were used in reproductive fitness and flight performance assays and were reared in individual families under the same density and diet conditions as parents (S1 Text).

For son reproductive fitness experiments, a total of 37 families were tested, with 19 in experiment one (five from converged and 14 from non-converged parents) and 18 in experiment two (10 converged and eight non-converged). For daughter fitness experiments, a total of 58 families were tested, with 25 in experiment one (six converged and 19 non-converged) and 33 in experiment two (16 converged and 17 non-converged). For son flight performance assays, a total of 51 families were tested, with 24 in experiment one (12 converged and 12 non-converged) and 27 in experiment two (seven converged and 20 non-converged).

## Reproductive fitness assays

**Male fertility: Insemination capacity.**   After recording their courtship flight interaction with a female, we assessed male fertility by measuring father and son insemination capacity, defined here as the proportion of females a male inseminated out of the total number of females presented to him over the course of the assay (S1 Text). Males were provided five new virgin females every other day for eight days (N = 20 total females presented per male), which represents their maximum number of potential mating partners during this time interval [12,44,46]. For fathers, a single male from each of the 57 unique families was tested, with 37 fathers tested in experiment one (13 converged and 24 non-converged) and 20 in experiment two (12 converged and eight non-converged). For sons, males from 37 families were tested (see previous section). Only males that survived the duration of the eight-day female insemination assay were included in the final analyses (see final sample sizes in Results section).

**Female fecundity: Eggs laid.**   For mothers, the number of eggs produced in her first clutch after her acoustic interaction and parental mating period was determined as described above. For daughters, reproductive potential was assessed using a horizontal life table approach as previously described [30] (S1 Text), recording blood feeding frequency, daily egg laying, and mortality for each female. A total of 214 females from 58 families were tested in daughter fitness experiments (see previous section).

## Flight performance assay

**Male flight performance: Mating attempts, contacts, and contact success rate.**   We measured the flight performance of fathers and sons using a custom-built flight response cage to test the ability of males to track, approach, and grasp a small moving speaker (Sony MH410C earphones, Sony Corporation, New York, NY, USA) emitting 550 Hz female-mimicking audio stimuli (S1 Fig and S1 Text). For fathers, flight performance was tested 24 h prior to testing for harmonic convergence. The flight performance protocol was developed through a series of pilot experiments and was chosen to be discerning (i.e., approximately 30% successful speaker contact rate) at the audio and speaker movement settings employed [47].

All flight performance trials were video recorded using a stationary high-definition camera (DCR-SR68, Sony Corporation, New York, NY, USA) positioned along the same horizontal axis as the moving speaker at a distance that allowed the camera to maintain focus for the entirety of the speaker's path (S1 Fig). This allowed us to watch videos at 0.125 speed (approximately 3 frames/s) for detailed analysis of male movement and interaction with the speaker. For each male we recorded the total number of attempts, or the orienting to and acceleration toward the moving speaker. For each attempt we noted whether the male's legs successfully contacted the speaker as well as how many times contact was accomplished in each attempt (for real-time examples of attempted contacts and contacts with the speaker, see S1 Video). We then used these data to calculate total contacts and the contact success rate, or the proportion of attempts that resulted in any number of contacts. We measured flight performance for a total of 84 fathers, with 42 in experiment one (18 converged and 24 non-converged) and 42 in experiment two (15 converged and 27 non-converged), and 157 sons, with 24 families in experiment one (12 converged and 12 non-converged) and 27 families in experiment two (7 converged and 20 non-converged), or an average of three sons tested per family.

## Statistical methods

All statistical analyses were performed using SPSS Statistics software (SPSS version 24, IBM Corp., Armonk, NY, USA) and R statistical software (version 3.6.3) [48] in RStudio (version 1.2.5033) [49] (S1 Text). R model packages used included "lme4" [50], "lmerTest" [51], "survival" [52,53], "coxme" [54], and "ggplot2" [55], as well as standard companion packages such as "car" [56] and "emmeans" [57] throughout. P-values <0.05 were considered statistically significant and were derived from minimal models, which included linear mixed effects models (LMM) [50,51,58], generalized linear mixed models (GZLMM) [50], linear models (LM) [50], and generalized linear models (GZLM) [50] (for full model details, tests to verify model assumptions, and detailed model statistics, see S1 Text and S1 Statistics). A one-sample Kolmogorov-Smirnov (KS) and an independent samples Kruskal–Wallis (KW) test were used to characterize and compare parent and offspring reproductive fitness and flight performance parameter distributions. Kaplan-Meier curves [52,53,59] and a Cox frailty model [54,57] were used to visually compare and test for differences in daughter survival based on parental convergence status.

Samples sizes for each experiment were determined beforehand to be optimal for subsequent analyses using standard statistical power analyses [60]. Data are listed as averages ± standard deviation (SD) throughout. Raw numerical data for all figures and tables are included in S1 Data.

## Results

### Harmonic convergence does not signal overall inherent quality

We first tested whether harmonic convergence functions as an acoustic signal of overall inherent quality in parents, as predicted in good genes selection models. No differences were detected in either father fertility (LM, P = 0.271; Fig 2A) or mother fecundity (LM, P = 0.245; Fig 2B) between parents that converged and those that did not. We next tested whether harmonic convergence signals indirect genetic benefits that accrue to offspring of converged parents. As with parental traits, neither son (LMM, P≥0.122 for both comparisons; Fig 2C) nor daughter (LMM, P≥0.331 for all comparisons; Figs 2D and 3A–3C) reproductive fitness measures, including daughter life table metrics (Fig 3D and S1 Table) and survival (Cox frailty model, P = 0.693; Fig 3E), differed by either offspring (in the case of sons) or parental convergence status (in the case of sons and daughters). Furthermore, sons of converged parents were no more likely to converge than sons of non-converged parents (GZLMM, P = 0.364; Fig 2E).

The ability not only to perceive, but also to localize and track moving females in flight is a key component of male mating behavior. However, as with reproductive fitness traits, we found no relationship between father convergence status and flight performance metrics (LM and GZLM, P≥0.370 for all comparisons; Fig 4A–4D). Likewise, we found no significant differences in these same flight metrics between sons descending from converged parents and those descending from non-converged parents (LMM and GZLMM, P≥0.202 for all comparisons; Fig 4E–4H). Together, these findings suggest that harmonic convergence signals neither

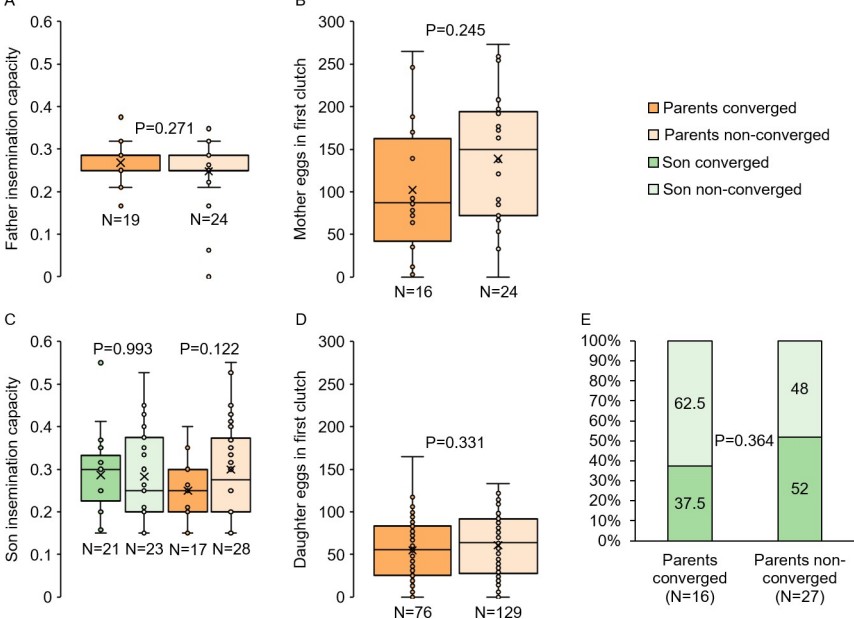

**Fig 2. Harmonic convergence signals neither parent nor offspring reproductive fitness traits.** Neither father fertility (A), mother fecundity (B), son fertility (C), nor daughter fecundity (D) differed by either parent or offspring harmonic convergence status (P≥0.122 for all comparisons). Son convergence status did not differ between those descending from converged parents and those descending from non-converged parents (E). Graphs display sample sizes (N) and LM, LMM, or GZLMM P-values for each comparison.

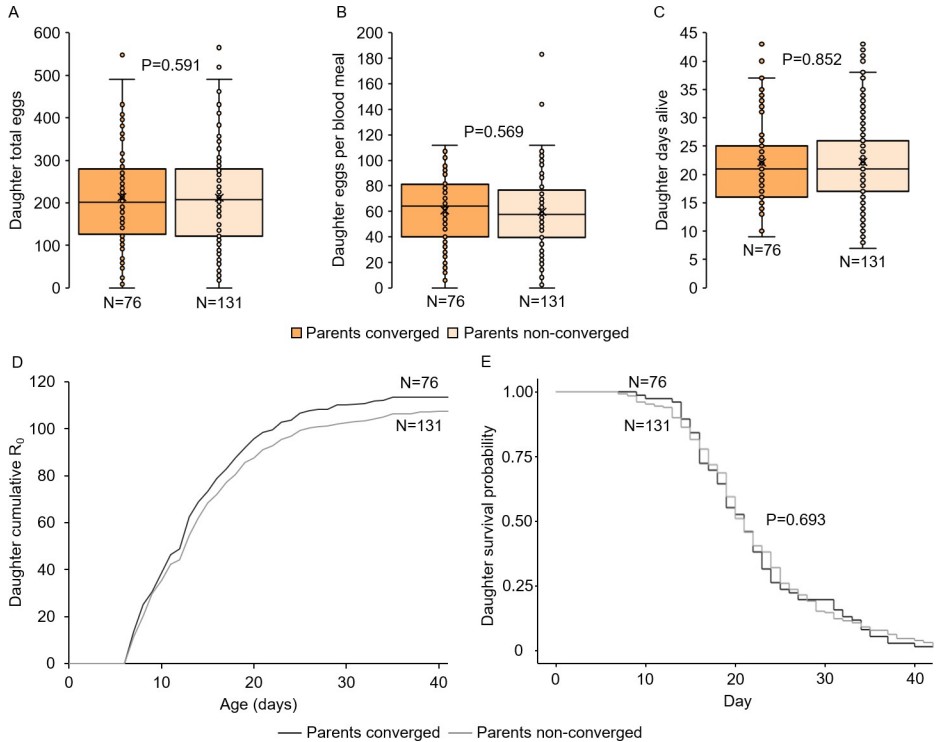

**Fig 3. Parental harmonic convergence does not signal daughter lifetime fecundity and life history parameters.**
Neither daughter fecundity (A and B) nor longevity (C) differed by parental harmonic convergence status (P≥0.569 for all comparisons). Daughter reproductive rate (D) and survival (E) also did not differ by parental convergence status. Graph displays sample size (N) and LMM (A–C) or Cox frailty model (E) P-values for each comparison. For details on daughter life table analysis data, see S1 Table. Survival data shows Kaplan-Meier curves for daughters across the duration of the life table study.

parent nor offspring inherent quality with respect to the reproductive fitness and flight performance metrics assayed here.

## Females that mate with higher quality males do not produce higher quality offspring

To determine whether females that mate with higher quality males produce higher quality offspring, we first measured the levels of standing variation in the fitness traits assayed here to assess the degree to which sexual selection may potentially occur in our low generation *Ae. aegypti* colony from Thailand. Overall, we detected significant standing variation in both reproductive fitness and flight performance trait outcomes both within and between parental and offspring generations (S1 Text and S2–S5 Figs and S2 and S3 Tables).

To test the possibility that females increase their offspring quality indirectly by mating with higher quality males, we examined the relationship between parent and offspring reproductive fitness and flight performance traits to assess their heritability. Overall, neither father nor mother reproductive fitness traits predicted offspring quality as measured by the same traits (LMM, P>0.05 for all comparisons; Figs 5 and 6 and S1 Statistics) with only one exception: larger females produced daughters that lived longer (LMM, P = 0.02; S1 Statistics). Furthermore, flight performance metrics of fathers displayed no correlation to that of their sons (LMM and GZLMM, P≥0.294 for all comparisons; Fig 7 and S1 Statistics). Thus, although the parent and offspring reproductive fitness and flight performance traits measured here were

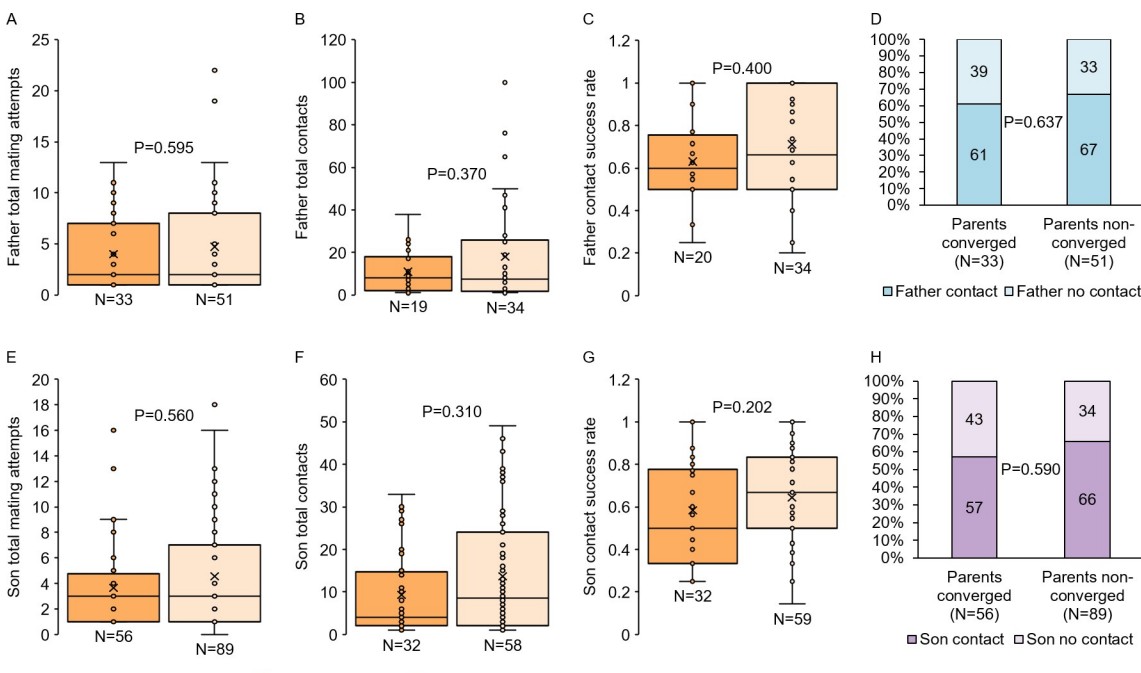

**Fig 4. Harmonic convergence signals neither parent nor offspring flight performance.** Neither father (A–D) nor son (E–H) flight performance metrics differed by parental convergence status (P≥0.202 for all comparisons). Graphs display sample sizes (N) and LMM or GZLMM P-values for each comparison.

highly variable, they were only weakly heritable. Hence, we find no evidence that females that mate with higher quality males reliably produce higher quality offspring as approximated by the traits assayed here.

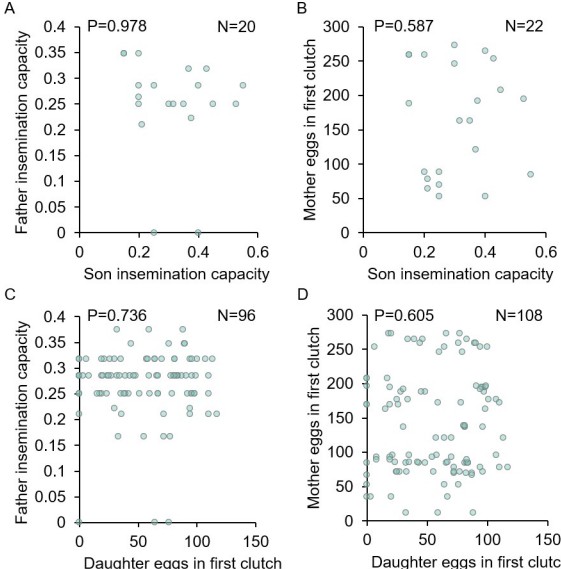

**Fig 5. Parent reproductive fitness traits display low heritability and do not correlate with offspring traits.** Father fertility and mother fecundity did not correlate with either son fertility (A and B) or daughter fecundity (C and D; P≥587 for all comparisons). Graphs display correlation sample sizes (N) and LMM P-values for each comparison. For detailed statistics, see S1 Statistics.

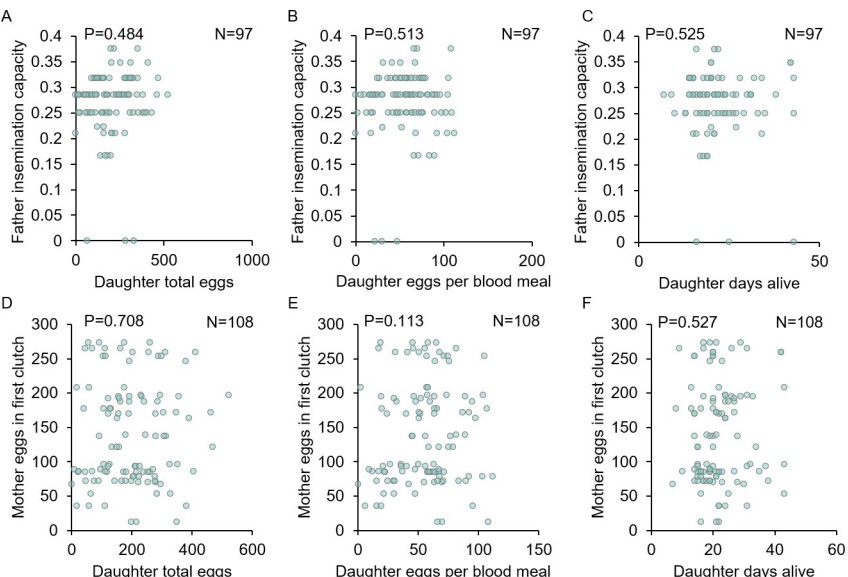

**Fig 6. Parent reproductive fitness traits do not correlate with daughter lifetime fecundity or life history traits.** Neither father fertility (A–C; P≥0.484 for all comparisons) nor mother fecundity (D–F; P≥0.113 for all comparisons) correlated with daughter fecundity or longevity. Graphs display correlation sample sizes (N) and LMM P-values for each comparison. For detailed statistics, see S1 Statistics.

## Discussion

Here, we tested the extent to which harmonic convergence serves as a good genes signal of overall heritable male quality. To do so, we surveyed the potential acoustic signaling, variability, and heritability of a broad array of male and female reproductive fitness and flight performance traits over multiple generations of a low generation mosquito colony originating from

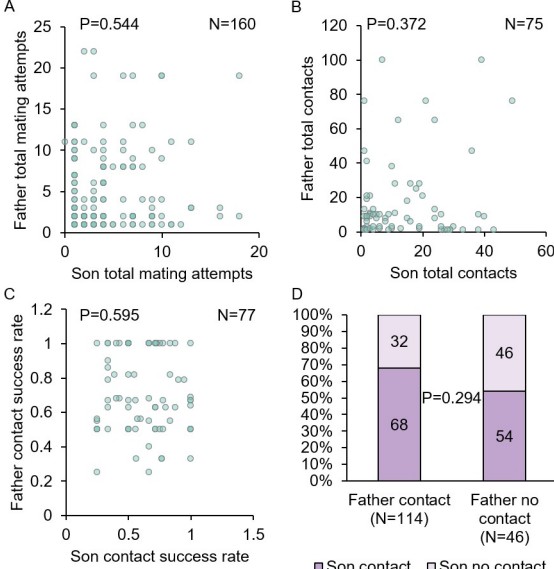

**Fig 7. Father flight performance displays low heritability.** Father flight performance metrics did not correlate with the same metrics in their sons (A–D; P≥0.294 for all comparisons). Graphs display correlation sample sizes (N) and LMM or GZLMM P-values for each comparison. For detailed statistics, see S1 Statistics.

dengue-endemic field sites in Thailand [61]. We found no evidence that harmonic convergence serves as a signal of male quality to females for the traits examined here. Furthermore, although the traits we examined displayed ample variation for sexual selection to act upon, parental traits displayed low heritability and did not correlate with offspring quality. Thus, the cumulative data presented in this study suggest that harmonic convergence does not serve as a good genes signal of inherent parent or offspring genetic quality with respect to these fitness measures.

We found no evidence that harmonic convergence serves as a signal to females of either inherent male quality or indirect benefits to offspring for the traits assayed here, as both parent and offspring fitness measures were identical regardless of convergence status (Figs 2–4 and S1 Table). Furthermore, in line with a previous study [19], as mother fitness was the same whether or not she converged with her mate (Fig 2B), females do not appear to receive direct benefits from harmonically converging males. Harmonic convergence is associated with mate recognition [62] and mating success [19,20,28] and has been documented across multiple mosquito species and genera [62–65]. Although father harmonic convergence was previously shown to signal son convergence and mating success [19], we did not observe this relationship in our present study (Fig 2E). The precise reasons for this difference are unclear. The Thai mosquitoes used in the present study were derived from wild populations that may have experienced different selective pressures that led to differences in mating behavior compared to the Mexican strains tested in the earlier study. This difference in mating behavior may also stem from the degree to which the strains examined experienced laboratory selection [43,66–68], as the earlier high generation strain had experienced more generations (7–8 generations) of laboratory selective conditions compared to the low generation strain used here. Importantly, while male mating success was assayed under competitive conditions in the previous work, male harmonic convergence and subsequent mating here occurred in one-on-one interactions, and a recent study in *Ae. aegypti* suggests that performance in these two conditions can differ [42]. Future work should address the variation in mating behavior between natural and laboratory adapted populations suggested by these findings.

In addition to potentially serving as a signal that females use to distinguish between males, harmonic convergence may occur as a consequence of the physical, aerodynamic coordination entailed in copula formation [20,69]. However, even if convergence is partly a byproduct of copula formation, our finding that males that successfully mate (i.e., inseminate females) do not sire offspring with higher mating success remains—converging males would still be more likely to form a mating copula than non-converging males. An additional consideration is the robustness of harmonic convergence as a readout of male mating success across multiple male-female encounters. Similar to previous studies, our study design classifies males as either converging or non-converging based on a single male-female courtship interaction. However, the repeatability of convergence events both within the same courting pair and between combinations of individuals across multiple interactions merits further investigation. Similar experiments could also be performed using the flight performance assay we developed here to determine the extent to which inter- and intra-male variation might impact this measure of male quality. Further characterization of potential sources of variation in harmonic convergence and male flight performance represents a crucial next step for understanding the biological significance of these behaviors and how they impact mating success. Finally, despite the advantages of examining harmonic convergence between free-flying males and tethered females [19,20,64], future studies building on previous work examining individual free-flying mosquito acoustics [70] could capture nuances of this behavior in free-flying couples that are not fully recapitulated using current standard methods.

Our data do not support a good genes model of indirect sexual selection for the fitness traits examined here. Previous studies have observed correlations between parental harmonic convergence and both offspring immunity [40] and mating success [19], consistent with good genes and Fisherian selection models, respectively. We therefore propose that sexual selection likely affects *Ae. aegypti* in a trait-, population-, and environment-dependent manner. Good genes models posit that females prefer males expressing traits that result from pleiotropic effects of genetic loci controlling multiple aspects of male fitness, such as those that specifically increase reproductive fitness [36,71], immune performance [72], or overall condition [73,74]. However in Fisherian models, a positive feedback loop between heritable female preference and arbitrary male traits leads to selection for both [75]. In this scenario, known as Fisher's runaway process [38,75], the traits which females find attractive are correlated with male attractiveness alone and have no relationship with other traits contributing to male fitness. Consistent with this, as well as our present findings, a meta-analysis of indirect benefits studies found that while offspring life history traits, such as the ones examined here, were not associated with father attractiveness, traits directly linked to paternal mating success were [76]. While we did measure a performance trait here (i.e., flight performance), most studies that have documented associations between offspring performance and father attractiveness were related to immunocompetence [40] and physical condition, for example as expressed in weight at maturity [76]. Future studies could further investigate these selection models and the extent of their applicability by assaying additional traits, conducting detailed pedigree analyses in various mosquito lines, or varying environmental conditions.

We detected robust variation in the parent and offspring reproductive fitness and flight performance traits measured here (S1 Text and S2–S4 Figs and S2 Table), suggesting ample variation for sexual selection to act upon. We endeavored to capture overall male quality by selecting traits that have established links to reproductive fitness in *Ae. aegypti* [19,20,30,41,42,77] and that require the coordination of multiple physiological systems. We also assessed lifetime fitness directly in daughters with a life table study (Fig 3D and S1 Table) and indirectly in males by measuring insemination capacity (Figs 2A and 2C and 5A–5C and S2 Table), which is a known predictor of lifetime fitness [41]. Although male flight performance metrics examined here have not been previously linked to lifetime fitness, we chose them as general indicators of male agility and reflexes and hypothesize that males with stronger flight performance are likely to mate with more females in competition with other males and thus sire more offspring. Furthermore, first egg clutch size in our daughters was predictive of lifetime egg laying (LMM, P = 0.001; S6 Fig), suggesting that this trait is also predictive of lifetime fitness in females. Given that we did not directly test field-collected adults, the extent to which the variation we observed in these traits represents standing variation retained from the field as opposed to increased variation due to a relaxation of selection pressures in the laboratory remains unknown [78]. Conversely, in spite of our use of a low generation field-derived colony, we cannot rule out countervailing effects on variation from bottlenecks, founder effects, and the early stages of laboratory colony adaptation, which can eventually cause divergent reproductive phenotypes and reduced fitness and genetic diversity in mosquitoes [66–68,79]. Future work involving multiple colonization events examined over additional generations and biological replicates could help to resolve the contributions of these potential effects.

By analyzing our reproductive fitness and flight performance traits by family, we were able, for the first time in a large-scale study, to directly compare parent and offspring fitness parameters. Our data show that the fitness traits examined here display low levels of heritability (Figs 5–7), suggesting that females cannot increase their offspring quality with respect to these traits by mating with higher quality fathers. This finding is consistent with an earlier meta-analysis that documented lower heritability in both vertebrate and invertebrate life history traits due to

their complex and pleiotropic nature compared to morphological traits [80]. This pattern is also suggestive of stabilizing selection [81], which would imply that these traits are regulated by trade-offs with other aspects of fitness, as has been suggested recently in both *Ae. aegypti* [42] and *Anopheles gambiae* [82]. However, we cannot rule out the possibility that other reproductive fitness traits not examined here, or perhaps the same traits examined under different environmental conditions [83], would show inheritance consistent with good genes models. We did detect one correlation between parent and offspring traits, namely, that larger mothers produced longer-lived daughters. Since larger females display increased teneral reserves and fecundity [84] and increased survivorship to adulthood in offspring has been linked to maternal effects [85], it is possible that these mothers provide additional resources to their daughters that improve their survival.

## Conclusions and implications for mosquito control

A growing number of mosquito control programs, such as those utilizing *Wolbachia* [10], gene drive [11], and other modified insect techniques [8,9], involve the mass-rearing and subsequent release of modified adult mosquitoes to alter mosquito populations. For these measures to succeed, it is critical that modified mosquitoes successfully compete with their wild counterparts for mates. Although harmonic convergence can serve as a helpful readout of mating success [19,20,28] and at least some measures of inherent quality [40], our results suggest that harmonic convergence is not a reliable indicator of all aspects of overall mosquito quality and that the relationship between some parental and offspring fitness measures is not easily predictable. Hence, care should be exercised when using harmonic convergence and the other fitness measures examined here to infer various aspects of modified mosquito strain quality relative to their natural counterparts [86]. This is particularly true of cases where potential impacts of natural, laboratory, or sexual selection upon modified lines are unknown. Control programs may therefore benefit from focusing on the maintenance of traits with established relationships to harmonic convergence and inherent fitness as well as those that directly correlate with increased mating success, keeping in mind that the latter may outweigh, to some extent, other potential fitness costs and that these relationships may vary in different contexts.

Our results also suggest that control programs should account for substantial variation in reproductive fitness and flight performance traits in low generation colonies, even under controlled laboratory rearing regimens. Ideally, such programs should aim to identify baseline natural variation in relevant fitness traits of target mosquito populations prior to release of their modified counterparts to better inform the design of equipment and rearing protocols for working with these populations at large scales in the laboratory. More broadly, the standing variation we observed in key fitness traits is likely maintained by natural selection in wild populations. Mass-rearing protocols that drastically reduce variation in fitness traits can have important effects on program implementation and success [15,87]. The application of sexual selection theory to mosquito disease vectors is a critical, yet often overlooked component of modern disease control programs [15]. Further work investigating the determinants of mating success in mosquitoes could aid in the design of novel vector control strategies as well as improve the sustainability of current methodologies.

## Supporting information

**S1 Text. Supporting Methods and Results.**
(DOCX)

**S1 Fig. Flight performance assay set up.** A custom-built flight cage was fit with a moving arm that allowed for playback of audio stimuli from a moving source. A motorized timing belt attached to a speaker (A) was used to move the arm back and forth at fixed speeds. A stationary HD camera was positioned along the same horizontal axis as the speaker movement to capture male responses in the flight cage (B). Photograph credit: LJC.
(TIF)

**S2 Fig. Parent and offspring reproductive fitness traits displayed widely variable outcomes both within and across a single generation.** Father and son fertility (A and B) as well as mother and daughter fecundity (C and D) varied substantially. Graphs display distribution sample sizes (N) and the number of samples per bin (above bars). For detailed descriptive statistics, see S2 Table.
(TIF)

**S3 Fig. Daughter lifetime fecundity and life history trait outcomes display high variation.** Daughter lifetime fecundity (A), fecundity by blood meal (B), and longevity (C) displayed high levels of variability. Graphs display distribution sample sizes (N) and the number of samples per bin (above bars). For detailed descriptive statistics, see S2 Table.
(TIF)

**S4 Fig. Father and son flight performance trait outcomes display robust variation.** Father and son total mating attempts (A and B), total contacts (C and D), and contact success rates (E and F) displayed strong levels of variation. Graphs display distribution sample sizes (N) and the number of samples per bin (above bars). For detailed descriptive statistics, see S2 Table.
(TIF)

**S5 Fig. Parent and offspring sizes displayed only moderate variation and were similar between generations.** Father and son sizes (A and B) were normally distributed and biologically comparable, despite differing statistically (KW test, P = 0.041). Mother sizes (C), but not daughter sizes (D), were normally distributed and size distributions were biologically similar, despite differing statistically (KW test, P = 0.004). Graphs display distribution sample sizes (N) and the number of samples per bin (above bars). For detailed descriptive statistics, see S3 Table.
(TIF)

**S6 Fig. Daughter first egg clutch size predicts lifetime fecundity.** Daughters that laid more eggs in their first clutch tended to lay more eggs across their lifetime (P = 0.001). Graphs display correlation sample size (N) and LMM P-value.
(TIF)

**S1 Video. Flight performance assay male-speaker interactions.**
(WMV)

**S1 Statistics. Model summaries for parent and offspring fitness comparisons.** The LMM (and GZLMM for the final comparison) model summaries presented here were run with replicate as a fixed effect and family as a random effect to account for repeated measures. Models are listed in order of appearance in the results. Figure references are placed next to corresponding statistical outputs when applicable. P-values ≤0.05 are listed in bold.
(XLSX)

**S1 Data. Raw numerical data for all figures and tables.**
(XLSX)

**S1 Table. Daughter life table parameters do not differ by parental harmonic convergence status.** Daughter life table analysis data presented by parental convergence status and experimental trial (individually and averaged). Although the effect of parental convergence status on $R_0$ and $r$ (but not $T_c$) are inconsistent across two experimental trials, the average effect was similar. Abbreviations: $R_0$, cumulative reproductive rate; $T_c$, generation time; $r$, intrinsic rate of increase.
(XLSX)

**S2 Table. Parent and offspring reproductive fitness and flight performance trait descriptive statics reveal ample standing variation for sexual selection.** Mother and daughter fecundity displayed higher levels of variability compared to father and son fertility. Daughter lifetime fecundity and life history traits, particularly those related to egg laying, displayed the highest levels of variability of any measured parent or offspring fitness parameter. Father and son flight performance traits also displayed robust variation. Normality values display one-sample KS test P-values. Abbreviations: Min, minimum value; Max, maximum value; SD, standard deviation; Var, variance; Sk, skewness.
(XLSX)

**S3 Table. Parent and offspring size descriptive statistics reveal only minor variation.** Both parent and offspring size tended to vary only moderately. Normality values display one-sample KS test P-values. Abbreviations: Min, minimum value; Max, maximum value; SD, standard deviation; Var, variance; Sk, skewness.
(XLSX)

## Acknowledgments

We thank Dr. Andrew Aldersley for feedback on acoustic analysis methods.

## Author Contributions

**Conceptualization:** Garrett P. League, Laura C. Harrington, Sylvie A. Pitcher, Courtney C. Murdock, Lauren J. Cator.

**Data curation:** Garrett P. League, Laura C. Harrington, Sylvie A. Pitcher, Julie K. Geyer, Lindsay L. Baxter, Julian Montijo, Lynn M. Johnson, Lauren J. Cator.

**Formal analysis:** Garrett P. League, Laura C. Harrington, Lynn M. Johnson, Lauren J. Cator.

**Funding acquisition:** Laura C. Harrington, Lauren J. Cator.

**Investigation:** Garrett P. League, Laura C. Harrington, Sylvie A. Pitcher, Julie K. Geyer, Lindsay L. Baxter, Julian Montijo, Lauren J. Cator.

**Methodology:** Garrett P. League, Laura C. Harrington, Sylvie A. Pitcher, Julie K. Geyer, Julian Montijo, John G. Rowland, Lynn M. Johnson, Lauren J. Cator.

**Project administration:** Garrett P. League, Laura C. Harrington, Sylvie A. Pitcher, Julie K. Geyer, Lindsay L. Baxter, Julian Montijo, Courtney C. Murdock, Lauren J. Cator.

**Resources:** Laura C. Harrington, John G. Rowland, Lauren J. Cator.

**Software:** Garrett P. League, Lynn M. Johnson, Lauren J. Cator.

**Supervision:** Garrett P. League, Laura C. Harrington, Sylvie A. Pitcher, Courtney C. Murdock, Lauren J. Cator.

**Validation:** Garrett P. League, Laura C. Harrington, Sylvie A. Pitcher, Lynn M. Johnson, Lauren J. Cator.

**Visualization:** Garrett P. League, Laura C. Harrington, Lauren J. Cator.

**Writing – original draft:** Garrett P. League, Laura C. Harrington, Lauren J. Cator.

**Writing – review & editing:** Garrett P. League, Laura C. Harrington, Sylvie A. Pitcher, Julie K. Geyer, Lindsay L. Baxter, Julian Montijo, John G. Rowland, Lynn M. Johnson, Courtney C. Murdock, Lauren J. Cator.

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
