## [Decision Letter · Decision Letter 0]

15 Mar 2021

Dear Dr. Cator,

Thank you very much for submitting your manuscript "Sexual selection theory meets disease vector control: Testing harmonic convergence as a “good genes” signal in Aedes aegypti mosquitoes" for consideration at PLOS Neglected Tropical Diseases. As with all papers reviewed by the journal, your manuscript was reviewed by members of the editorial board and by several independent reviewers. In light of the reviews (below this email), we would like to invite the resubmission of a significantly-revised version that takes into account the reviewers' comments. 

We cannot make any decision about publication until we have seen the revised manuscript and your response to the reviewers' comments. Your revised manuscript is also likely to be sent to reviewers for further evaluation.

Sincerely,

Pattamaporn Kittayapong, Ph.D.

Associate Editor

Karin Kirchgatter, Ph.D.

Deputy Editor

Reviewer's Responses to Questions

**Key Review Criteria Required for Acceptance?**

**Methods**

-Are the objectives of the study clearly articulated with a clear testable hypothesis stated?

-Is the study design appropriate to address the stated objectives?

-Is the population clearly described and appropriate for the hypothesis being tested?

-Is the sample size sufficient to ensure adequate power to address the hypothesis being tested?

-Were correct statistical analysis used to support conclusions?

-Are there concerns about ethical or regulatory requirements being met?

Reviewer #1: (No Response)

Reviewer #2: Line 191: Unclear. By having ‘, or’ are your describing the male fertility trials or describing a second type of assay?

Line 195: unclear. Was it a single male from each of 57 unique families?

Line 230: I’m unclear about the appropriateness of the distributions in some of the models used. It seems as though LMMs were applied to count data which were not normally distributed for example, table 1 states that data for most assays were not normally distributed, but then figure 4/supp info states that LMMs were used for some of those assays. For example, to me “father/son total mating attempts” or “father/son total contacts” seem to be count data best modelled using GLMMs with Pois/negbin distributions not LM/LMMs. Is that not correct? 

Additionally, as your detailed explanations are far away in the supp info, perhaps adding the model type/distribution/link function information to somewhere more easily accessible up front like table 1 or I guess even the s1 tables may help readers understand exactly what models you ran for each assay and how appropriate they were. 

Line 820: please provide speaker spec details

Line 832: unclear. How long did it take the speaker to go from one side of the cage to the other? 

Line 838: ‘the through” typo

Line 862: why would males not be successful at contacting the 200 RPM speaker when they could get stationary, 300 & 400 RPM do you think? 

Line 929 (S1 fig): I’m unclear about the camera in the set up. Did it swing back and forth at the same speed of the speaker on a separate timing belt? Is that what you mean by parallel to speaker movement? If it was to the side of the box as the line figure depicts how did it maintain focus? I would really love to see a photo of the entire unit if you have it rather than the line box drawing.

Reviewer #3: The main objective of this study is to test if the acoustic behavior known as harmonic convergence can signal heritable male quality. This objective is based on hypotheses of indirect sexual selection, which assume correlations between sexually selected traits and offspring fitness. The authors explored two predictions: (i) females use harmonic convergence as a signal to detect good genes and (ii) females that mate with higher quality males produce higher quality offspring. In addition, the authors designed and implemented an innovative methodology to evaluate male flight performance as a trait associated with fitness. Although the experimental approach, population, sample size and statistical analyses implemented in the study are sufficient to test hypotheses associated with male fertility and female fecundity, I found the experiments used to test flight performance misleading. 

Flight performance assays were performed by testing each male only once in a 2-minute trial. This experiment assume that all males have the same motivation to locate a sound source. However, this might not be the case. If the variables used to assess flight performance are directly correlated to mating success, for instance, males should exhibit similar performance in several trials. If this is not the case, the experiment is affected by other factors, for instance, by how motivated individual males were during the trial. In addition, the experiment does not show if there are statistical differences on flight performance among males. I am not sure that the experimental design provides data to evaluate traits directly related to mating success. Finally, the authors acknowledge that unlike male fertility and female fecundity, flight performance has not been evaluated as a fitness trait in mosquitoes before. For these reasons, conclusions based on this trait that might be misleading.

**Results**

-Does the analysis presented match the analysis plan?

-Are the results clearly and completely presented?

-Are the figures (Tables, Images) of sufficient quality for clarity?

Reviewer #1: (No Response)

Reviewer #2: Line 326: I’d insert wing length here rather than just size as we don’t know how you measured size until line 760.

Reviewer #3: The large amount of work reflected by the results is remarkable. Two main findings are presented in the manuscript. First, harmonic convergence does not signal male quality. Second, quality traits examined in this study displayed low heritability. The way results are presented to the reader make this section easy to follow. On the contrary, some figures are hard to read due to the big amount of information condensed in several panels which contain, in many cases, redundant titles and legends. Also, Table 1 does not provide useful information in the main text. I suggest moving that table to supplemental information. On the contrary showing that parental harmonic convergence does not signal offspring convergence is an important result as this finding challenges previous studies. I suggest including Supplemental Figure S3 in the manuscript.

**Conclusions**

-Are the conclusions supported by the data presented?

-Are the limitations of analysis clearly described?

-Do the authors discuss how these data can be helpful to advance our understanding of the topic under study?

-Is public health relevance addressed?

Reviewer #1: (No Response)

Reviewer #2: Line 428: "measures measured" seems like an odd word choice.

Line 464: Does your work support the idea that potentially “harmonic convergence is a by-product of the physical (e.g. aerodynamic) coordination” (Andres et al. 2020)? With all the words on harmonic convergence and sexual selection, I think that this alternate notion needs be addressed somewhere in your manuscript, if not the conclusion section.

Reviewer #3: The authors did a great job at making strong cases to support the main conclusions. However, I found the flight performance assay problematic. Evaluating flight performance as a trait directly linked to paternal mating success is innovative, however, the absence of replicates proving statistical evidence of differences among individuals might lead to incorrect conclusions. Based on the methods it is unclear if flight performance is associated with traits, like flight agility or reflexes, that directly correlate with mating success. Testing if a male shows consistent flight performance in several replicates would make a stronger case to support these conclusions. 

Although most analyses limitations are acknowledged and discussed by the authors, I find concerning the use of tethered females to test sexual selection. Free-flying females react to approaching males by modulating their flight-tones (1,2). Mosquito courtship is therefore a complex acoustic interaction affected by mosquito motion. Harmonic convergence assays, however, were performed by using “semi-tethered” females. Although this approach has been used in the past to evaluate harmonic convergence and it allows some flight mobility, the female’s response is still constrained. This limitation might affect selection and should be acknowledged by the authors.

1. Simões PM V. et al. 2016 A role for acoustic distortion in novel rapid frequency modulation behaviour in free-flying male mosquitoes. J. Exp. Biol. 219, 2039–2047

Minor comment

The authors found that father harmonic convergence does not signal son ability to converge. This finding is relevant as challenges previous studies and promote future research. Although it is acknowledged that the specific reason for this difference is unclear, the authors speculate that differences in selective pressures might led to differences in mating behavior between strains used in this and previous studies. Were differences on behavior detected? Is it possible to support this hypothesis using collected data?

**Editorial and Data Presentation Modifications?**

Reviewer #1: (No Response)

Reviewer #2: Abstract: I don’t think it is good practice to include the importance of findings of ‘previous work’ in the summary statement of your abstract. I’d suggest re-writing lines 39- and focus on the importance of the work specific to this study. 

Introduction: 

Line 94 felt a bit broad with the mention of nutritional gifts and parental effort. Are there examples of direct material benefits relating to mosquitoes. If not, maybe state this. If so, it would be interesting to learn about them here. 

Line 103: should ‘as’ be ‘are’?

Reviewer #3: Overall, figures are the main weakness of the manuscript from the editorial perspective. In general, figures condense a lot of information which in many cases is redundant. Although the general idea is comprehensible, reducing the number of legends and titles will improve the figures considerably. In addition, it is necessary to increase letter size and resolution of all figures. 

- In Figure 1, it is necessary to differentiate harmonic convergence from the traits that were tested as indicators of quality. As it is, Figure 1 can be misinterpreted. Using different geometric figures or labels to what shapes and colors mean will make the figure more intelligible.

- In Figures 2, 3 and 4, labels are redundant. For instance, the authors could add only one label in the figure to indicate convergence for all the panels. 

- Increase letter size in Figure 3.

- The X axis of panels E and F are hard to read in Figure 7. Please increase space between numbers. 

- Increase resolution of all figures.

**Summary and General Comments**

Reviewer #1: League et al present a study on the importance of harmonic convergence of mosquito wingbeats on mating success and life history traits. The mating behavior of Aedes aegypti is not as well studied as that of the malaria mosquito or the fly, so this is a welcome area. In this work, the authors were surprised to find no benefit (or effect at all) of harmonic convergence on mating attempts, mating success, or overall fecundity/life history of their offspring. This is consistent with some of their previous work, such as a 2011 paper where they also found no effect on life history traits. However, unlike in that work, they report here that convergence itself was not a heritable trait in the particular strain used. This indicates that there may be regional differences in selection pressures that lead to harmonic convergence. However, as the authors note, the two studies are not directly comparable due to differences in experimental design (competitive vs non-competitive matings), so it is difficult to draw firm conclusions in either direction. This inability to firmly reject or confirm previous findings is a major limitation of the work. The presence of other confounding effects such as artefacts of colonization further diminish the generalizability of the results, limiting impact. While the presentation of the manuscript is generally well-written, there are some redundant aspects that can be streamlined that would improve the clarity of the presentation. 

The authors do not specify, but it appears that matings were given unlimited time. One possible advantage of harmonic convergence is to shorten the mating time, an advantage for females and their future offspring in the sense that they can more rapidly return to a safe harborage. However, in the absence of predators this advantage disappears. 

An assumption of this work is that males can be accurately classified as “converging” or “non-converging” based on a single attempt with a randomly selected female. Is there evidence that males that display convergence will always display it, and vice versa? If there is sufficient variance in female flight tones, it is not unreasonable to think that males can better match some tones but not others. 

For measurement of flight traits, no indication is provided as to standardization of time of day, as flight behavior is strongly controlled by the circadian clock. 

Line 81: “Mating in Aedes mosquitoes occurs rapidly (<60 s) in aerial swarms [16] composed primarily of males and only a small number of females [17,18].”

This statement is misleading and misrepresents what is given in the provided references. For example, in REF 17, the authors of that study describe that mating in Aedes can occur in swarms, but also in small groups or individually, citing evidence for each. 

Lin147: “low generation” means what, exactly? The authors should be more precise here since this is a subjective term. 

Lin150: experiments were conducted on adults 2-6 days post emergence. I worry about this range. 2-day old adults are likely not able to display full feeding/host seeking behaviors. This could be an additional source of confounding variation in the data. 

Lin160: Authors should describe what their criteria for “definitively rejected” were. The methods in their 2011 paper were much more precise, maybe cite that here if the same criteria were used.

Lin163: please clarify whether “replicated twice” means “performed twice” or “performed three times” (an initial experiment and two replicate experiments). From the follow-up text, it appears the experiment was performed twice, which means it was replicated only once. 

Lin177: after bloodmeal was any indication used to confirm that all mosquitoes fed to repletion? Differences in life history traits could be due to differences in the amount of blood imbibed.

Line 282: : another reference to “low generation” of colony. 

Fig 5-7. These figures are just a re-expression of the same data from Figs2-4, looking at total variation. These are not needed, as the data is already presented and the normality/non-normality/range of the data is already visible. 

The data is presented a 3rd time as Table 1. This can probably be moved to the supplement, as it does not add anything beyond what is already presented. 

Lin324-325. This drop in egg numbers between mothers and daughters is concerning. Even if the trait itself were not highly heritable, I would expect the overall range to be similar. This suggests problems in colonization which the authors allude to briefly in the discussion and make it difficult to generalize these data. 

Line 442: The authors are commended for discussing the limitations of the study, but their phrasing “we cannot entirely rule out…” gives the impression that they can mostly rule these out, when in fact that is not the case. Ruling these out would require multiple colonization events and substantially more replication. Quite the opposite, as the small scale of this study (only two replicates over three generations) renders it particularly susceptible to the whims of these confounding effects.

Reviewer #2: The researchers performed a range of laboratory trials to test the influence on harmonic convergence on multiple fitness traits of Aedes aegypti and did not find any relationships. The work was very well executed, analysed and written. I really enjoyed reading the paper and only have a few minor comments which hopefully may help to further improve this work.

Reviewer #3: This study constitutes an important piece to move forward in our understanding of mosquito reproduction. In the manuscript, the authors address challenging questions by performing a large-scale study that investigates how acoustic communication mediate sexual selection in mosquitoes. The findings presented in the manuscript are relevant as they not only provide new information but challenge previous findings. 

My main concern about this manuscript is that flight performance assays might not capture what the authors intended. The idea is very creative and might promote future research. However, proving that the flight performance assays capture traits directly associated with mating success is necessary. Would it be possible to perform a short experiment testing flight performance of the same males during two or three days? If not, it would be necessary to acknowledge this limitation in the discussion or explaining why the authors predicted that males will exhibit the same flight performance during harmonic convergence or insemination assays. 

A secondary concern is that figures condense a large amount of data, which in some cases makes them difficult to read. In addition, many figures contain redundant information. Reducing the number of labels and titles can make the figures clearer for the reader. Also, improving the resolution, size letter and space between numbers in the axis will make the manuscript more attractive and intelligible for a general audience.

PLOS authors have the option to publish the peer review history of their article (what does this mean?). If published, this will include your full peer review and any attached files.

Reviewer #1: No

Reviewer #2: No

Reviewer #3: No
---

## [Decision Letter · Decision Letter 1]

4 Jun 2021

Dear Dr. Cator,

We are pleased to inform you that your manuscript ' Sexual selection theory meets disease vector control: Testing harmonic convergence as a “good genes” signal in Aedes aegypti mosquitoes ' has been provisionally accepted for publication in PLOS Neglected Tropical Diseases.

Best regards,

Pattamaporn Kittayapong, Ph.D.

Associate Editor

Karin Kirchgatter, Ph.D.

Deputy Editor

Reviewer's Responses to Questions

**Key Review Criteria Required for Acceptance?**

**Methods**

-Are the objectives of the study clearly articulated with a clear testable hypothesis stated?

-Is the study design appropriate to address the stated objectives?

-Is the population clearly described and appropriate for the hypothesis being tested?

-Is the sample size sufficient to ensure adequate power to address the hypothesis being tested?

-Were correct statistical analysis used to support conclusions?

-Are there concerns about ethical or regulatory requirements being met?

Reviewer #1: (No Response)

Reviewer #2: All my points were thoroughly addressed. I look forward to reading the published version. Well done.

Reviewer #3: (No Response)

**Results**

-Does the analysis presented match the analysis plan?

-Are the results clearly and completely presented?

-Are the figures (Tables, Images) of sufficient quality for clarity?

Reviewer #1: (No Response)

Reviewer #2: (No Response)

Reviewer #3: (No Response)

**Conclusions**

-Are the conclusions supported by the data presented?

-Are the limitations of analysis clearly described?

-Do the authors discuss how these data can be helpful to advance our understanding of the topic under study?

-Is public health relevance addressed?

Reviewer #1: (No Response)

Reviewer #2: (No Response)

Reviewer #3: (No Response)

**Editorial and Data Presentation Modifications?**

Reviewer #1: (No Response)

Reviewer #2: (No Response)

Reviewer #3: (No Response)

**Summary and General Comments**

Reviewer #1: The authors have responded positively the all of the comments, and have carefully revised the manuscript to provide critical experimental details/methods, improved clarity in the figures, and appropriate acknowledgement of remaining uncertainty regarding the limits of their study. Overall this is a substantial amount of work that contributes to our knowledge of Aedes mating behaviour, with important implications for pest management approaches that rely on mating success.

Reviewer #2: (No Response)

Reviewer #3: The authors have satisfactorily addressed all my comments and made changes that improve the manuscript.

I appreciate the clarification regarding the flight performance experiment. As the authors mention in their response, my concerns about this particular assay are similar to the concerns that reviewer 1 has about harmonic convergence. Although the authors have added a paragraph (lines 386–391) to address these comments, I encourage the authors to further discuss how intra-male variation might affect this kind of experiment. Discussing this idea might help the reader to design novel experiments and move forward in this research field.

PLOS authors have the option to publish the peer review history of their article (what does this mean?). If published, this will include your full peer review and any attached files.

Reviewer #1: No

Reviewer #2: No

Reviewer #3: No

---

## [Editor Report · Acceptance letter]

28 Jun 2021

Dear Dr. Cator,

We are delighted to inform you that your manuscript, " Sexual selection theory meets disease vector control: Testing harmonic convergence as a “good genes” signal in Aedes aegypti mosquitoes ," has been formally accepted for publication in PLOS Neglected Tropical Diseases.

Best regards,

Shaden Kamhawi

co-Editor-in-Chief

Paul Brindley

co-Editor-in-Chief
